**Subject Category:**
Biology (whole organism)

evolution

geometric morphometrics, intraspecific variation, macaques, mandible, phenotypic plasticity, primates

**Author for correspondence:**
Tsuyoshi Ito
e-mail: ito.tsuyoshi.3a@kyoto-u.ac.jp

# Phenotypic plasticity in the mandibular morphology of Japanese macaques: captive–wild comparison

Siti Norsyuhada Kamaluddin[1], Mikiko Tanaka[2], Hikaru Wakamori[2], Takeshi Nishimura[2] and Tsuyoshi Ito[2]

[1]School of Environmental and Natural Resource Sciences, Faculty of Science and Technology, Universiti Kebangsaan Malaysia, 43600 Bangi, Selangor, Malaysia
[2]Department of Evolution and Phylogeny, Primate Research Institute, Kyoto University, Inuyama, Aichi 484-8506, Japan

 TI, 0000-0001-6193-2408

Despite the accumulating evidence suggesting the importance of phenotypic plasticity in diversification and adaptation, little is known about plastic variation in primate skulls. The present study evaluated the plastic variation of the mandible in Japanese macaques by comparing wild and captive specimens. The results showed that captive individuals are square-jawed with relatively longer tooth rows than wild individuals. We also found that this shape change resembles the sexual dimorphism, indicating that the mandibles of captive individuals are to some extent masculinized. By contrast, the mandible morphology was not clearly explained by ecogeographical factors. These findings suggest the possibility that perturbations in the social environment in captivity and resulting changes of androgenic hormones may have influenced the development of mandible shape. As the high plasticity of social properties is well known in wild primates, social environment may cause the inter- and intra-population diversity of skull morphology, even in the wild. The captive–wild morphological difference detected in this study, however, can also be possibly formed by other untested sources of variation (e.g. inter-population genetic variation), and therefore this hypothesis should be validated further.

## 1. Introduction

Phenotypic plasticity refers to the ability of a single genotype to produce distinct phenotypes in response to varying environmental conditions [1]. Cursory consideration would suggest that plasticity

has no relevance to evolutionary processes other than dampening the effects of selection. However, theoretical and empirical studies have suggested that plasticity could play a key role in promoting diversification at numerous levels, often through the mechanism of genetic accommodation or more specific genetic assimilation [1–6]. In other words, plasticity can promote the emergence of novel phenotypes, diversification within and among populations and species, and adaptive radiation [1]. In fact, it has been reported that clades that exhibit more ecologically relevant plasticity are more species-rich and have broader geographical ranges than closely related clades lacking plasticity [7]. Therefore, evaluating phenotypic plasticity, not only genetically determined variation, is important to understand the evolutionary processes of diversification and adaptation.

Comparison between captive and wild individuals provides clues to interpret phenotypic plasticity, even for taxa in which experiments are difficult or impossible due to ethical and/or physical constraints (most intermediate- and large-bodied mammals are included in this group). Although the purposes of such research have varied, many scholars have examined whether there are morphological differences between captive and wild skeletal specimens in various mammalian taxa [8–24]. Studies on skulls have often focused on the effects of mechanical loading associated with dietary differences. For example, food is usually softer in a captive environment than in the wild. Possibly as a result of this dietary shift, captive individuals tend to exhibit a less developed sagittal crest, less doming of the dorsal roof of the skull, and greater zygomatic and/or palatal widths than wild ones, in several Carnivora species (i.e. lions [20], tigers [9], and coyotes [17]). The studies of squirrel monkeys also showed the dental anomalies and narrower dental arches in the individuals raised on soft rather than hard diets [23]. These findings imply that the relaxation of mechanical loading in captivity probably results in anomalies and less development in the regions that are susceptible to masticatory stress.

Social animals, such as some primates, can also be influenced by the skewed social environment encountered in captivity. Although this perspective has been largely overlooked in the literature, Singleton [24] pointed out the potential influences of social environment on skull development. By experimental simulation, she demonstrated that the characteristics of captive male *Mandrillus*, for example, robust cranial superstructures, increased facial height and length, and facial retroflexion, can be mostly explained by extension of the normal male developmental trajectory [24]. The background to this phenomenon lies in the roles of androgenic hormones, which directly and indirectly influence the development of skulls, particularly during adolescence [25–28]. The level of testosterone (one of the androgenic hormones) is associated with social rank or dominance behaviour [29–31], and such social properties are to some extent changed by captive environments [32–36]. For example, it is plausible that captive males, who typically lack or have a limited number of male peers [37], experience unchallenged dominance, and therefore are exposed to prolonged, elevated testosterone levels. Thus, the masculinized skulls often observed in the captive individuals of social primates are likely to be at least in part attributable to perturbation of the social environment and resulting changes in socio-hormone levels.

As one of social primates, there has been a relatively large amount of literature on Japanese macaques, also known as snow monkeys, regarding their ecology, society, genetics and morphology, and a variety of laboratory studies since the establishment of the field of primatology in Japan in 1948 [38]. This species is distributed across a wide climatic range spanning from subtropical to cool temperate, the northern limit of which is the highest inhabited latitude among the living species of non-human primates [39]. Some scholars have investigated the ecogeographical variations of morphology in Japanese macaques [40–45]. These works revealed that some traits, such as body size [45], nasal cavity size [41] and relative and absolute molar sizes [44], exhibit a latitudinal cline or a negative correlation with temperature; these geographical clines may directly or indirectly reflect adaptation to climatic gradient. One of the plausible factors indirectly causing this is the regional difference in dietary composition, which is correlated with climatic conditions; for example, individuals in colder environments and/or higher latitudes more frequently eat bark, buds, and herbaceous plants in winter [46–48]. It is possible that the masticatory apparatus has adapted to such regional differences in dietary conditions. On the other hand, non-metric skull traits appear to reflect genetic variations of blood protein polymorphisms [49]. Thus, the morphological variations in Japanese macaques have been relatively well documented and discussed in the context of adaptive or neutral evolution. However, despite its importance in the evolutionary context as mentioned above, little attention has been paid to phenotypic plasticity in morphological traits.

In the present study, we investigated phenotypic plasticity in mandibular morphology in Japanese macaques as potentially influenced by dietary composition and/or androgenic hormone level during development. In particular, we evaluated how, and to what extent, captivity influences mandibular

**Table 1.** Samples used in this study. The number in parentheses indicates young adult specimens.

| prefecture | population | environment | latitude | longitude | female | male | total | |
|---|---|---|---|---|---|---|---|---|
| Aomori | Shimokita | wild | 41.51 | 140.93 | 4 | 4 | 8 | (0) |
| Miyagi | Kinkazan | wild | 38.29 | 141.57 | 5 | 2 | 7 | (0) |
| Toyama | Hakusan | wild | 36.29 | 136.64 | 9 | 9 | 18 | (0) |
| Saitama | Nagatoro | captive | 36.11 | 139.11 | 1 | 1 | 2 | (0) |
| Nagano | Kamimatsu | wild | 35.78 | 137.69 | 1 | 0 | 1 | (0) |
| Nagano | Matsukawa | wild | 35.6 | 137.91 | 0 | 1 | 1 | (0) |
| Nagano | Takamori | wild | 35.55 | 137.88 | 1 | 0 | 1 | (0) |
| Fukui | Takahama | captive | 35.49 | 135.55 | 19 | 13 | 32 | (10) |
| Fukui | WakasaF | wild | 35.49 | 135.75 | 6 | 6 | 12 | (3) |
| Aichi | Inuyama | captive | 35.38 | 136.94 | 1 | 0 | 1 | (0) |
| Shimane | WakasaS | captive | 35.34 | 134.4 | 12 | 9 | 21 | (0) |
| Shimane | Mitoya | wild | 35.3 | 132.89 | 1 | 0 | 1 | (0) |
| Shimane | Kotsugu | wild | 35.29 | 132.9 | 1 | 1 | 2 | (1) |
| Shimane | Yoshida | wild | 35.17 | 132.85 | 0 | 1 | 1 | (0) |
| Kyoto | Arashiyama | captive | 35.01 | 135.67 | 9 | 4 | 13 | (2) |
| Shiga | Koga | wild | 34.97 | 136.17 | 10 | 5 | 15 | (1) |
| Shimane | Hasumi | wild | 34.87 | 132.62 | 4 | 4 | 8 | (2) |
| Shizuoka | Izu | captive | 34.86 | 138.94 | 4 | 1 | 5 | (1) |
| Osaka | Minoo | captive | 34.85 | 135.47 | 1 | 1 | 2 | (0) |
| Kagawa | Shodoshima | captive | 34.51 | 134.3 | 1 | 0 | 1 | (0) |
| Hiroshima | Miyajima | captive | 34.28 | 132.31 | 4 | 5 | 9 | (0) |
| Wakayama | Noguchi | wild | 33.9 | 135.18 | 0 | 1 | 1 | (0) |
| Kagoshima | Yakushima | captive/wild | 30.29 | 130.44 | 7 | 8 | 15 | (5) |

morphology, by comparing captive to wild specimens. We then compared the direction of captivity-related shape change with the directions of shape change related to other factors, namely, size, sex, age class and ecogeographical factors as a proxy for dietary conditions. To achieve these, we applied a Bayesian mixed model approach because Bayesian inference facilitates the intuitive interpretation of results compared with classical frequentist inference [50] and because a mixed model can take into account unknown random effects regarding population differences [51].

## 2. Material and Methods

The sample analysed in this study consists of 177 skeletal specimens of Japanese macaques, housed at the Hakusan Nature Conservation Centre, Hakusan, and the Primate Research Institute (PRI), Kyoto University, Inuyama, Japan (table 1). Among these, 93 were captive individuals, while 84 were wild. Captive individuals had been raised at PRI, for the purpose of research and/or breeding. Of the 93 captive individuals, 39 were founders (coded as zero generation), which had been transferred from 10 different natural provisioned or non-provisioned populations to PRI (figure 1). The other 54 captive individuals were the first to the fourth generation derived from such founders. The captive individuals had usually been fed monkey chow (AS: Oriental Yeast Co., Ltd.) and sweet potato. They had been housed in an outdoor enclosure, a group or isolated cage; they had sometimes been transferred among these three places for research, clinical care or breeding management. The 84 wild specimens were derived from 14 populations, ranging from subtropical Yakushima to cold-temperate Shimokita (figure 1). The wild specimens were derived from cadavers collected in the wild or those obtained from government management of populations. No animals were sacrificed for the purposes of the present study. To rule out developmental variations, the sample only consists of the mature

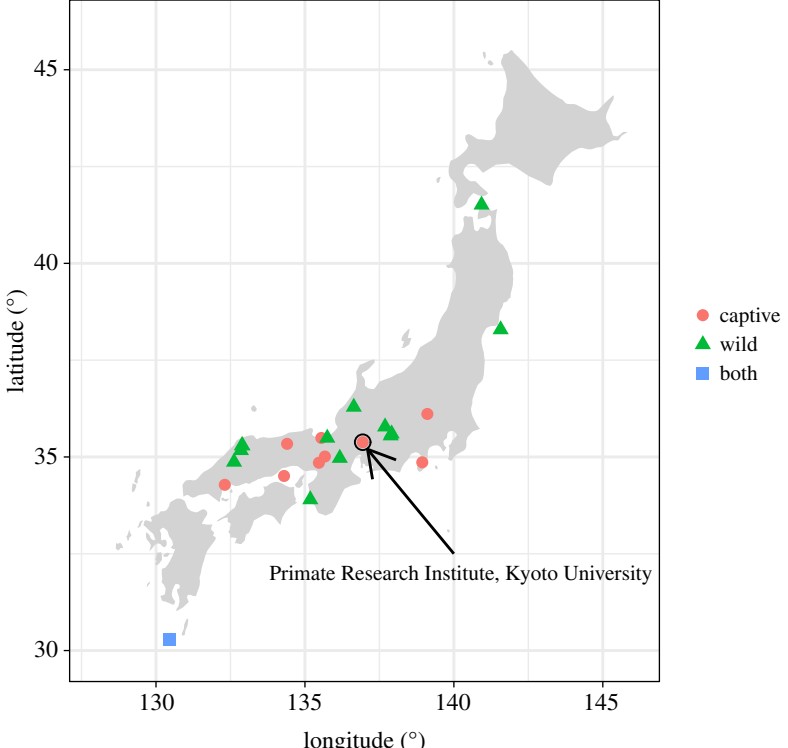

**Figure 1.** Localities of samples. Red circles denote the localities where the founders of the captive individuals were captured. Green triangles indicate the localities where wild individuals or their remains were collected. A blue square denotes a locality from which both captive and wild specimens were derived.

specimens with almost (young adult, $N = 25$) or fully erupted third molars (adult, $N = 152$). Age ranges from 6.4 to 29.5 years (mean $\pm$ s.d.: $14.0 \pm 5.7$) for captive specimens, although it is unknown for wild specimens. Fully erupted third molars (adults) were observed in individuals with 8.1 years of age or more. They include both sexes (female, $N = 101$; male, $N = 76$). No specimen exhibiting any evidence of severe alveolar pyorrhoea was included in the samples.

The mandibles were scanned using a computed tomography (CT) scanner (Latheta LCT-200; Hitachi-Aloka Ltd., Tokyo, Japan) at the Institute for Genetic Medicine, Hokkaido University, with a slice thickness of 0.24 mm; another CT scanner (Asteion Premium 4; Toshiba Medical Systems Co., Otawara, Japan) at PRI, with a slice thickness of 0.5 mm; or a 3D laser scanner (NextEngine Inc., Santa Monica, CA, USA) at PRI. Twenty-eight landmarks (table 2 and figure 2) were digitized by a single observer to avoid inter-observer error and were then double-checked by another, using Stratovan Checkpoint software (Stratovan Co., Sacramento, CA, USA).

Geometric morphometrics [52,53] was applied to evaluate variations in the shape and size of the mandibles, using Morpho [54] and geomorph [55] packages, and a custom script written in R software [56]. In advance of the analyses, one outlier was detected and removed based on the Smirnov–Grubbs test ($p = 0.05$) of the Procrustes distance from a mean shape. The generalized Procrustes analysis was performed to superimpose landmark configurations. The symmetric shape components were subjected to principal component analysis to summarize shape variations. Because shape components are redundant and often highly correlated [57], the number of variables used for the analyses of shape was reduced by including only the first 10 principal components (PCs). The first 10 PCs accounted for 80.3% of total variance, and their pairwise Euclidean distances were highly correlated with the matrix of Procrustes distance ($r = 0.98$), indicating a good summary of shape variations (electronic supplementary material, figure S1). The natural logarithm of centroid size was calculated and used as a measure of size in the following analyses (hereinafter referred to as 'size').

A Bayesian linear mixed model was applied to evaluate factors influencing mandible shape and size, using the brms package [58,59] in R. Here, the response variable was shape or size. For shape, a multivariate response model was applied, wherein PC1–10 were simultaneously used as response variables. In the test of the effect of captivity, size (if response was shape), sex, age class and captivity were used as fixed effects (table 3). Sex was coded as 0 (female) or 1 (male); age class as 0 (young

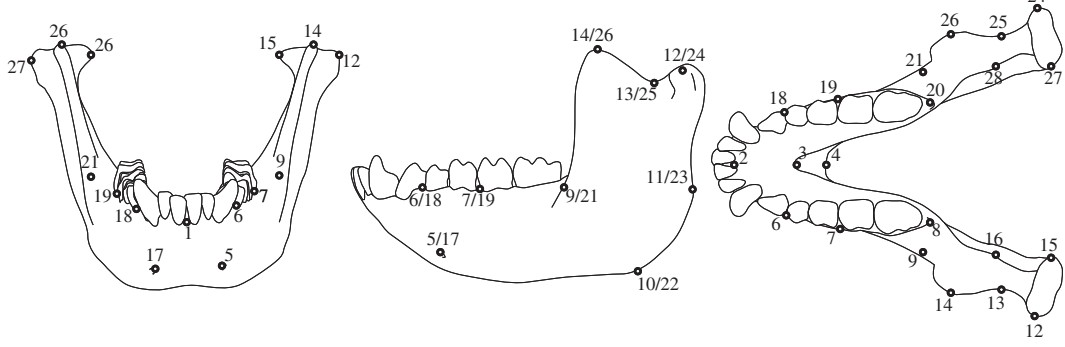

**Figure 2.** Landmarks used in this study. Frontal (left), left-lateral (middle) and occlusal (right) views of mandibles.

**Table 2.** Landmarks used in this study.

| no | | landmark |
| --- | --- | --- |
| 1 | | infradentale |
| 2 | | mandibular orale |
| 3 | | superior transverse torus (posterior) |
| 4 | | gnathion |
| 5 | 17 | mental foramen (anterior) |
| 6 | 18 | P3 – P4 (lateral) |
| 7 | 19 | M1 – M2 (lateral) |
| 8 | 20 | M3 (lateral-posterior) |
| 9 | 21 | ramus (anterior and in line with alveolus) |
| 10 | 22 | gonion |
| 11 | 23 | ramus (posterior and in line with alveolus) |
| 12 | 24 | condylion laterale |
| 13 | 25 | sigmoid notch |
| 14 | 26 | coronion |
| 15 | 27 | condylion mediale |
| 16 | 28 | mandibular foramen (superior) |

adult with almost erupted third molars) or 1 (adult with fully erupted third molars); and captivity as 0 (wild), 1 (founder, i.e. zero generation), or 2 (first to fourth generations). The test of the effect of ecogeographical factors was performed using the subset of the wild-caught specimens; size (if response was shape), sex, age class and the annual mean temperature and annual precipitation for the past 30 years (1970–2000) were used as fixed effects. The two ecogeographical variables were obtained from the WorldClim database, using the raster package [60] in R (electronic supplementary material, figure S2). Latitude and longitude were not included in the model to avoid redundancy, as they were highly correlated with the two ecogeographical variables (absolute $r = 0.72–0.82$). For all of the models, population was set as a random effect to account for unknown variations regarding population differences. Before running the models, all variables were scaled to facilitate understanding of the results. We used an improper flat prior for fixed effects and a weakly informative prior (a half Student's $t$ prior with three degrees of freedom) for random effects, following the default settings of brms. Family, that is the distribution of response variable, was assumed to be Gaussian. The models were run with four MCMC chains, each of which had 10 000 iterations with 5000 burn-in, yielding a total posterior sample size of 20 000. We confirmed that Markov chains reached convergence (Rhat < 1.1).

The validity of the models was evaluated based on the widely applicable Bayesian information criterion (WAIC) [61], wherein the full models were compared with the reduced models that excluded the focused-on effects, namely, captivity or the two ecogeographical variables (table 3). WAIC is a

**Table 3.** Summary of models. ΔWAIC is calculated compared with the best model, which has the smallest WAIC. A blank in the ΔWAIC column indicates that it is the best model. All the models have random effect of population.

| explanatory variables (fixed effects) | Bayes $R^2$ | WAIC | s.e. | ΔWAIC | s.e. |
|---|---|---|---|---|---|
| *test of captivity for size* | | | | | |
| sex + age class + captivity | 0.720 | 296.4 | 27.9 | 2.0 | 0.5 |
| sex + age class | 0.721 | 294.4 | 27.8 | | |
| *test of captivity for shape* | | | | | |
| size + sex + age class + captivity | 0.327 | 4474.6 | 75.6 | | |
| size + sex + age class | 0.318 | 4484.3 | 74.6 | 9.7 | 10.2 |
| *test of ecogeographical factors for size* | | | | | |
| sex + age class + temperature + precipitation | 0.760 | 131.9 | 16.9 | 1.3 | 1.0 |
| sex + age class + precipitation | 0.761 | 130.6 | 17.1 | | |
| sex + age class + temperature | 0.757 | 132.0 | 17.0 | 1.4 | 2.0 |
| sex + age class | 0.758 | 131.4 | 17.4 | 0.7 | 1.7 |
| *test of ecogeographical factors for shape* | | | | | |
| size + sex + age class + temperature + precipitation | 0.322 | 2204.3 | 39.3 | 7.2 | 8.5 |
| size + sex + age class + precipitation | 0.311 | 2199.0 | 39.4 | 1.9 | 5.6 |
| size + sex + age class + temperature | 0.310 | 2204.4 | 38.7 | 7.3 | 5.1 |
| size + sex + age class | 0.299 | 2197.1 | 39.1 | | |

Bayesian analogue of Akaike's information criterion [62], which evaluates the goodness of a model (a smaller value is better). This was performed using the brms package.

The direction of captivity-related shape change was compared with each of those related to other fixed effects. This was performed by calculating the angles and correlations between the pairs of vectors of regression coefficients [63,64]. In the case of comparing vectors between different models, the vector of each iteration for a model was compared with the mean of vectors of another model; the same was done in the opposite direction, and the two sets of angles or correlations were combined. In addition, to illustrate the captivity-related shape change, we calculated a shape score for captivity. Shape score $s$ was calculated as follows: $s = y\beta^T(\beta\beta^T)^{-0.5}$, where $y$ is the matrix of shape variables and $\beta$ is the regression vector [65]. These calculations were performed using a custom script written in R [54,55,63,64].

To evaluate the validity of these analyses, we reproduced the Bayesian linear mixed models with changing variables or using the subset of samples as follows. First, to make doubly sure about the control of allometric effect, we replaced the response variables with the PCs of size-adjusted shape data. The size-adjusted shape data was calculated as the residuals from the multiple regression of the symmetric shape components on size using MorphoJ software [66] ($R^2 = 0.023$, $p = 0.048$). Second, to take island effect into account, we added island, which was coded as 0 (non-island) or 1 (island), to explanatory factor (fixed effect). Third, to consider the possible difference in age distribution between captive and wild samples, we reproduced the models by using the subset of samples that excludes the captive specimens with more than 20 years of age ($N = 167$). This was done because such old-aged individuals are considered to be rare in the wild [39]. Fourth, the models were reproduced including only adult (fully erupted third molars) specimens ($N = 151$).

## 3. Results

The captive environment did not have a significant effect on size or the majority of shape variations (figure 3 and electronic supplementary material, figure S3). Size was instead influenced by sex (0.76 [95% CI 0.68–0.84]) and age class (0.21 [0.13–0.30]), indicating that the mandibles of males and adults were larger than those of females and young adults, respectively. PC1 accounting for 22.0% of total variance explained the relative width of the mandible and the relative sizes of body and ramus (figure 4a). As the PC1 score decreased, size became larger (−0.67 [−0.83 to −0.51]). These findings

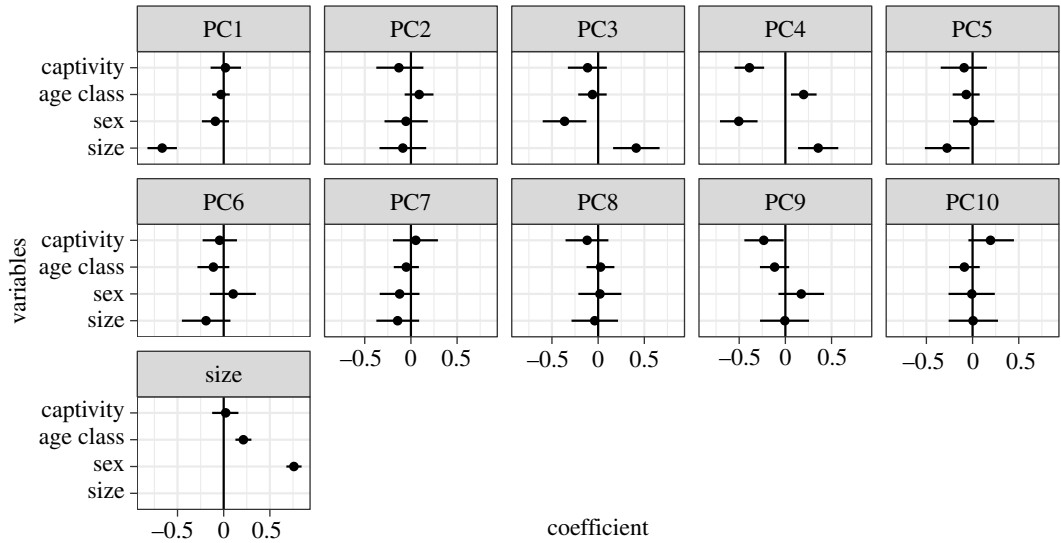

**Figure 3.** Posterior distributions of regression coefficients in the test of captivity. Points are means and lines represent 95% credible intervals.

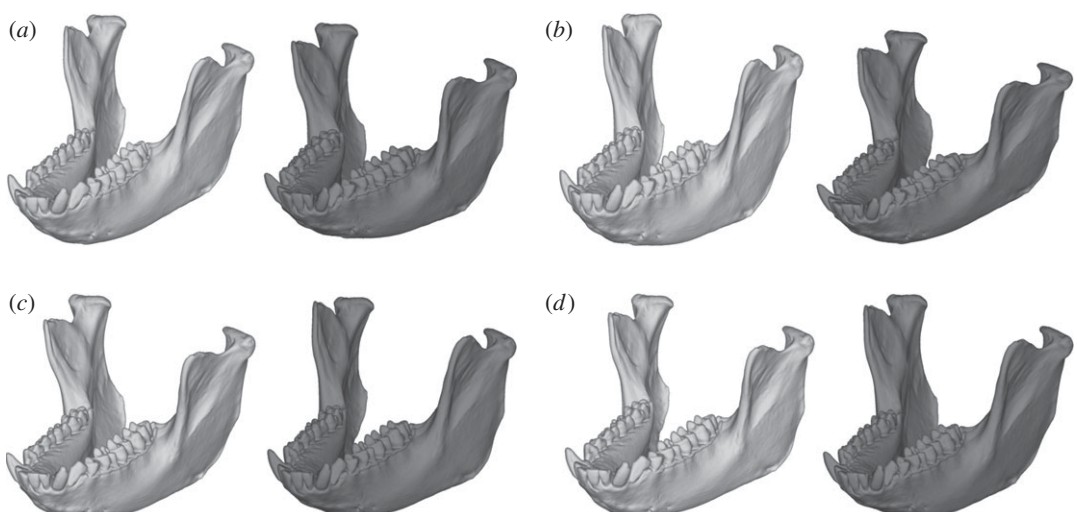

**Figure 4.** Shape changes along with each principal component axis. (*a*) PC1, (*b*) PC2, (*c*) PC3 and (*d*) PC4. The 3D models of negative (−3 s.d., left, white) and positive (+3 s.d., right, grey) extremes are shown.

indicate that, in larger individuals, the mandible is narrower and the mandibular body larger relative to the ramus. PC2 was explained by the relative height of the ramus but is not associated with any explanatory variables (figure 4*b*). PC3 (10.4%) score was significantly positively correlated with size (0.41 [0.16–0.67]) and negatively correlated with sex (−0.36 [−0.60 to −0.13]). This may be the consequence of multicollinearity between size and sex, because the two variables are highly correlated to each other ($r = 0.75$) and because, when removing one of the two variables, the effect of the other became insignificant (electronic supplementary material, figures S4 and S5). The individuals with larger PC3 scores tend to be square-jawed (laterally positioned gonions) with a more posteriorly inclined ramus (figure 4*c*). PC5–10 were not significantly explained by any explanatory factors.

PC4 (9.4%) was significantly influenced by captivity (−0.39 [−0.55 to −0.23]) as well as by age class (0.20 [0.06–0.34]), sex (−0.50 [−0.71 to −0.30]) and size (0.36 [0.14–0.57]) (figures 4*d* and 5). As PC4 score decreases (i.e. in captive rather than wild animals, in males rather than females, in young adults rather than adults and in smaller individuals), the tooth row becomes relatively longer, the gonion is positioned more laterally (being square-jawed), the ramus inclines more posteriorly with respect to the body, and the anterior part of the body is relatively more robust (figures 4*d* and 5). When viewing the shape variation in its entirety, we confirmed that the full model was much better than the reduced

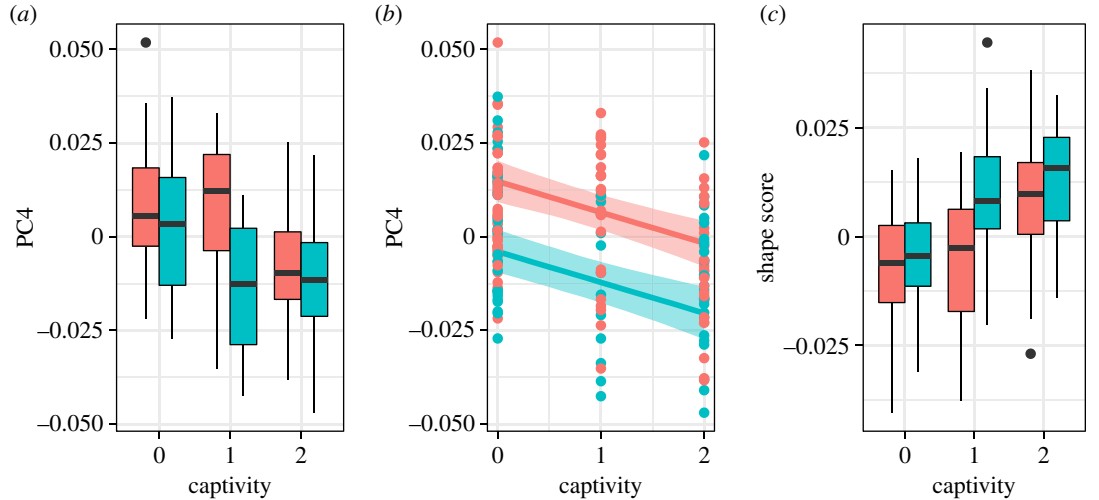

**Figure 5.** Relationship between mandible shape and captivity. Red indicates females and blue, males. (*a*) Boxplot between PC4 and captivity; (*b*) fitted values of the marginal effect of captivity on PC4; (*c*) boxplot between shape score and captivity.

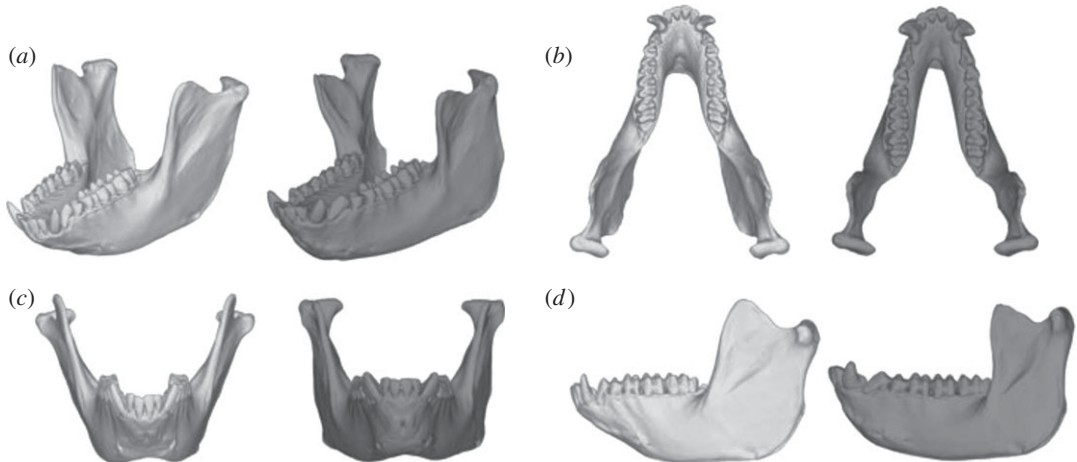

**Figure 6.** Shape change along with shape score. (*a*) Overview, (*b*) occlusal, (*c*) frontal and (*d*) lateral views of the 3D model. Negative (left, white) and positive (right, grey) extremes are shown. Note that the magnitude of shape change is exaggerated tenfold for clarity.

model, in which captivity is excluded from among the explanatory variables (table 3). The shape score was highly negatively correlated with PC4 ($r = -0.81$, $p < 2.2 \times 10^{-16}$), and the shape change with captivity was almost the same as (but in the opposite direction to) that along PC4 (figure 6); this indicates that shape changes with captivity are mostly represented by PC4. The small improvement in Bayes $R^2$, however, indicated that the effect of captivity on mandible shape explained a small proportion of the total variance (table 3).

Significant effect of captivity was detected even when using the PCs of size-adjusted shape data for response variables (electronic supplementary material, figure S6), when including island as explanatory factor (electronic supplementary material, figure S7), when using the subset of samples that excludes the captive specimens with more than 20 years of age (electronic supplementary material, figure S8), or that only consists of adult specimens (electronic supplementary material, figure S9). In the captive specimens with known age and generation, neither age nor generation was significantly correlated with PC4 and shape score (electronic supplementary material, table S1).

Figure 7 illustrates the vector angles and correlations between the effects of the different variables on mandible shape. Angles between the same pair of effects were similar, irrespective of whether comparison was performed between models or within a model, indicating that the difference in models did not markedly affect the effects of each factor on shape. The 95% credible intervals crossed the line representing independence (90° for angle and zero for correlation), except for the pair between sex and

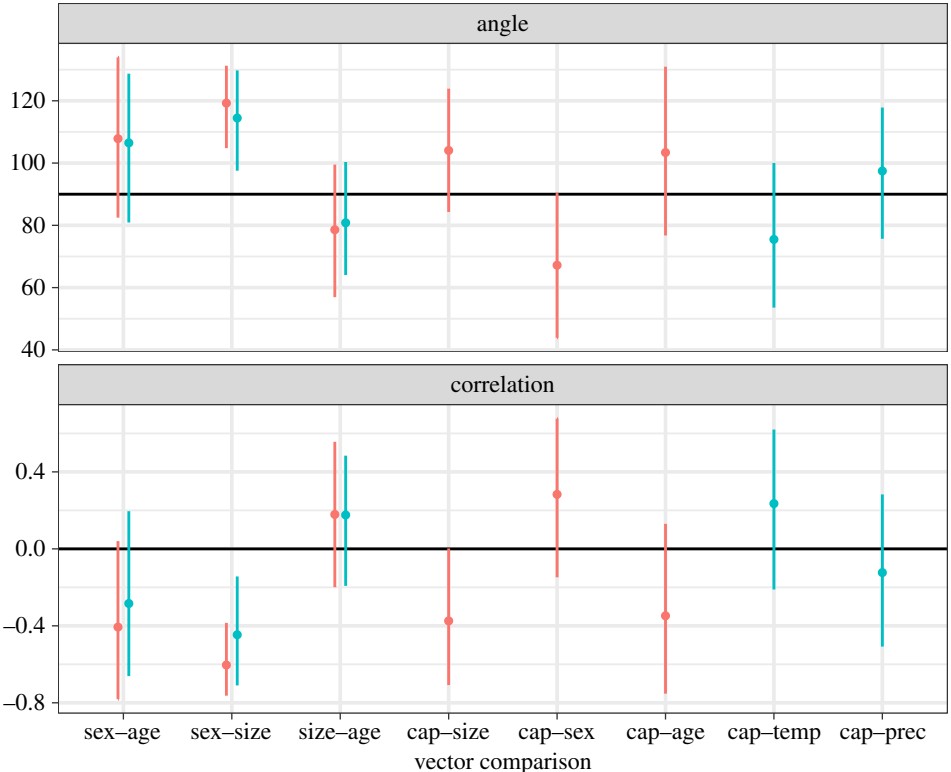

**Figure 7.** Vector angles and correlations between effects on shape. Red indicates the comparison between the effects within the same model. Blue indicates the comparison between the effects in different models. Points are means and lines represent 95% credible intervals. Solid line denotes independence (90° for angle and zero for correlation).

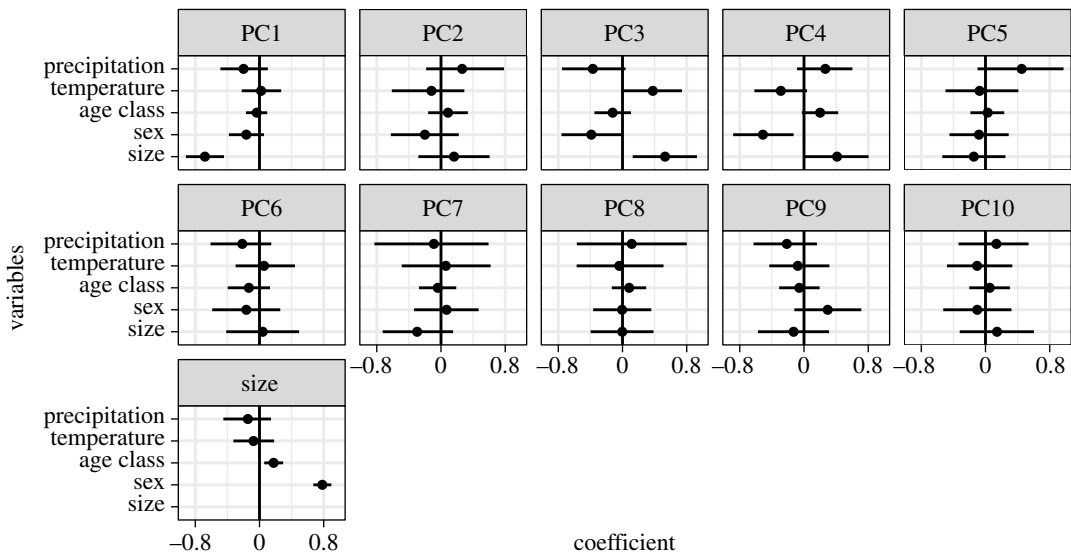

**Figure 8.** Posterior distributions of regression coefficients in the test of ecogeographic factors. Points are means and lines represent 95% credible intervals.

size. This indicates that, when viewing shape variation in its entirety, the direction of captivity-related shape change was not significantly related to any of the other factors.

For ecogeographical factors, temperature, not precipitation, was significantly associated with PC3 (0.38 [0.01–0.73]) (figure 8). This indicates that the individuals inhabiting colder environments tended to be square-jawed with a more posteriorly inclined ramus (figure 4c). However, when viewing the whole shape variation, the inclusion of ecogeographical factors considerably worsened the model (table 3 and electronic supplementary material, figure S10). In the model that considers island effect,

the model that includes only temperature was slightly better than the reduced one that excludes both temperature and precipitation (electronic supplementary material, figure S11). Thus, temperature might possibly have an effect but it is tentative.

## 4. Discussion

Size of the mandible was found not to differ between captive and wild Japanese macaques. This is consistent with some previous studies of various mammalian taxa that reported no differences in skull or body size between the two groups [17,67], although some studies reported decreased [8,15] or increased [15,68] overall skull size in captive individuals. This background suggests that the response of body size to captivity may be taxon-specific, and that the nutritional conditions in the captive environment could affect body size. The finding of the present study suggests that, in Japanese macaques, nutritional conditions in captivity were unlikely to have been markedly different from natural conditions. Size is instead best explained by age class and sex; that is, males and adults tend to be larger than females and young adults, respectively. Somewhat unexpectedly, size was found to be independent of both temperature and precipitation in wild populations. In primates, body or skull size is often positively correlated with precipitation in tropical regions (possibly reflecting the responses to the primary productivity of plants, and hence food availability) [69–72] or negatively with temperature in temperate regions (as predicted by Bergmann's rule) [73]. By contrast, the geographical variation of mandible size in Japanese macaques appears to reflect unknown factors, such as population history, which was incorporated into random effects, rather than local adaptation.

Major variations in shape also did not differ between captive and wild individuals. PC1 explains allometric shape variations, indicating that larger individuals tend to show narrower and more elongated mandibles. This seems to follow the common allometric trend of mammals, reflecting the truncation or extension of ontogenetic trajectory; larger animals tend to have longer faces/muzzles [74]. PC2 representing the relative ramus height cannot be explained by any apparent factor. Significant effects of size and sex on PC3 were detected, but this may be the consequence of multicollinearity. PC5 and subsequent PCs seem to be almost independent of the factors that we tested.

The present study also suggested that no shape variation is primarily explained by either temperature or precipitation, although temperature might possibly have an effect. Japanese macaques as a species have specialized mandibular morphology, which is considered to be advantageous for eating tough food such as bark and mature leaves, compared with other species of the same genus [75]; however, the intraspecific variation in mandible shape might not to be the consequence of dietary adaptation to local environments, despite the wide distribution range from subtropical to cool-temperate regions and related wide dietary variations. Although indication of dietary adaptation was detected in an analysis of the dental morphology in local populations of Japanese macaques [44], mandible shape, as with size, may be likely to reflect population history or other unknown factors rather than current climatic (temperature and precipitation) conditions.

A part of the shape variation differs between captive and wild individuals, which is mostly represented by PC4. Captive individuals tend to be square-jawed with relatively longer tooth row than wild ones. Because it is unlikely that the size of each tooth changes, this shape change likely reflects the change in the proportion of body and ramus and/or in the interval between teeth. This shape change resembles sex differences, which indicates that captive individuals exhibit a mandible shape that to some extent exaggerates male-like features, namely, being masculinized. These findings are consistent with the hypothesis that the change in mandible shape in captive individuals is attributable to perturbations in the social environment and resulting changes of androgenic hormones [24]. It should be, however, kept in mind that captivity explains a small proportion of total shape variance, and that the direction of captivity-related shape change is not identical to that of sex-related shape change, when viewing whole shape variations. In addition, the possibility of changes in mechanical loading having had an effect cannot be ruled out because we did not directly test the dietary shift but just evaluated the effects of ecogeographical variations as a proxy for dietary differences. Therefore, in Japanese macaques, the social environment in captivity possibly influences the development of mandible shape via changes in sex hormones, although this is not the only factor causing captivity-induced shape change, and its effect is small.

The captive–wild difference should be interpreted with care, because there are several other possible sources of variations, which were considered here. First, although this has been often overlooked in captive–wild comparisons, it is possible that differences in age distribution between captive and wild

samples influence morphology. The present study, however, detected significant captive–wild difference even when removing the old-aged captive specimens or when restricting the sample to adults (i.e. specimens with fully erupted third molars). We also did not detect significant shape change (in PC4 and shape score) with ageing after the full eruption of third molars. It was reported by van Minh *et al.* [76] that after maturation (seven years of age) the ramus inclines posteriorly and the overall dimensions of mandible increase with age in Japanese macaques, but such features are not observed in captive specimens when compared with wild ones. Therefore, at least in our sample, the age distribution difference is unlikely to explain any difference between captives and wilds. Second, it should be noted with care that genetic relationships between the study individuals might be a source for morphological variation, which we did not account for here. Although the present study partly controlled inter-population variations by incorporating population as a random effect in the mixed model or by evaluating the island–non-island difference, this cannot completely rule out the effect of genetic inheritance. Unfortunately, little is known about the nuclear genetic variation in wild populations of Japanese macaques so far, and further studies are needed to elucidate this issue.

In conclusion, this study suggested the possibility that captivity caused a masculinized feature in mandible shape possibly because of the perturbations in the social environments and resulting changes of androgen hormones, although this hypothesis should be validated further. If this is the case, social plasticity may have played a small but significant role in shaping the intra- and inter-population diversification of skull morphology in primates.

Ethics. This work was conducted with the skeletal specimens of non-human primates, and thus there were no ethical issues. No animals were sacrificed for the purposes of this study. None of the authors conducted any actions on the living animals that became the skeletal specimens.

Data accessibility. The datasets and code used in this study are available from the Dryad Digital Repository at: https://doi.org/10.5061/dryad.768vg4f [77].

Authors' contributions. S.N.K. collected the landmark data; M.T. and H.W. took the CT and 3D scanner data; T.N. was involved in the research design; T.I. was involved in the research design, took the CT data, analysed the data, and wrote the manuscript. All authors gave final approval for publication.

Competing interests. We declare we have no competing interests.

Funding. This work was partly funded by the Keihanshin Consortium for Fostering the Next Generation of Global Leaders in Research (K-CONNEX to T.I.) and the JSPS Grants-in-Aid for Scientific Research (grant nos 16H04848 and 19H01002 to T.N. and 17K15195 and 19K16211 to T.I.).

Acknowledgements. We are grateful to Hitoshi Fukase for sharing the CT data. We thank Tetsu Hayashi, Kosei Izawa, the study material committee of PRI, and all the persons who were involved in collecting and preserving the skeletal remains of Japanese macaques. We thank Naoko Suda–Hashimoto and Mayumi Morimoto for providing the information about the captive individuals and their inhabiting environments at PRI. We thank the laboratory members of the evolutionary morphology section of PRI for their fruitful comments on the early version of this study. We also thank two anonymous reviewers for their constructive comments that greatly improved our manuscript.

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
