## [Reviewer comments · Royal Society Open Science]

Review History

RSOS-181382.R0 (Original submission)

Review form: Reviewer 1 (Pasquale Raia)

Is the manuscript scientifically sound in its present form?

Yes

Are the interpretations and conclusions justified by the results?

Yes

Is the language acceptable?

Yes

Is it clear how to access all supporting data?

Yes

Do you have any ethical concerns with this paper?

No

Have you any concerns about statistical analyses in this paper?

No

Recommendation?

Major revision is needed (please make suggestions in comments)

Comments to the Author(s)

The paper is a sound, statistically good paper blessed with a good dataset. Although my general opinion is good, I'm worried by the excessive confidence of the authors in interpreting some results. I therefore suggest the authors to tone down some such insights, since they appear standing on thin ice at the moment.

More to the point:

1 You simply cannot control for genetic proximity in wild-caught individuals. How many of them a young males? do Japanese macaques distribute philopatrically? were sick or old individuals easier to collect? is there any pregnant female in the wild-caught sample? Please notice this matters to the evaluation of ecogeographical determinants of body size and body shape variation. Your result is perfectly reasonable, but all too many potential factors are simply ignored

2 at reading the intro I had the impression that sociality would be carefully scored and evaluated, but neither of the two. I wonder whether sociality fits anywhere in this manuscript, since it is either ignored (wild-caughts) or unnatural (captives)

3 the feeding status of captive individuals might be known, but did it vary seasonally? in wild individuals outside the tropics seasonal variation is the norm

4 being captive probably might elicit reaction norms of some sort, but most phenotypic variation is known to occur with domestic individuals, not wild-caught or F4 offspring of wild-caughts, so what is the basic premise of the ms? You have nice data there and a good treatment, and a nice paper. But I wonder if a better paper might be produced relaxing the basic expectations and premises (in the intro sections) and narrowing the scope (less sociality, less ecogeography, even less diet). I'd rather preserve these factors for the discussion, and produce a skinner, more frank presentation.

Thanks for the opportunity to go through this!

Pas

Review form: Reviewer 2

Is the manuscript scientifically sound in its present form?

Yes

Are the interpretations and conclusions justified by the results?

No

Is the language acceptable?

Yes

Is it clear how to access all supporting data?

Yes

Do you have any ethical concerns with this paper?

No

Have you any concerns about statistical analyses in this paper?

No

Recommendation?

Major revision is needed (please make suggestions in comments)

Comments to the Author(s)

The manuscript deals with an interesting question, namely the impact of captive lifestyle on the facial morphology of wild animals, here Japanese macaques. The 3D morphometric approach is up-to-date and appropriate. Based on their analyses, the authors conclude that captive lifestyle had an important impact on facial shape, presumably because of hormonal changes in conditions where inter-individual relationships are departing from the wild. This may be an interesting result that would anyway deserve to be more thoroughly discussed in the background of what is known in captive representatives of other species (and that is clearly and with extensive citations presented in the introduction). Unfortunately, some drawbacks limit, in the present form, the support of the data to this interpretation.

First, a basic presentation of the results, with a PCA, a bgPCA or a CVA, and a plot of the size across the different populations, would help a lot the reader to get into the data. It would for instance show rather immediately the amount of variation related to the captive lifestyle compared to the variation in the wild. What is the amount of allometry in the shape variation? Second, the analyses with the Bayesian models seem quite sophisticated, but they may suffer of problems in their design. Mandible change with food consistency has been widely documented, and it seems rather surprising that the authors do not find any such trend across the wild populations. What if they analyze wild populations alone? And if they analyze size-corrected data? Their sampling further includes small islets, where an insular syndrome could develop. They do not mention such potential sources of variations: as a basis for all other analyses, this has to be clarify.

Even more importantly, the authors only mention to have assessed the age issue by separating young (third molar not erupted) and adults (third molar erupted). They do not state when this event occurs in the animal's life. It seems anyway a quite basic way to deal with age, in a species which is presumably rather long lived. Bone remodeling in late life can be quite extensive, and occur following other directions than the changes occurring in early life (see Swiderski & Zelditch 2013). Therefore, more details and a better control for age is an absolute requirement for assessing the validity of all results and interpretations: what is the life expectancy of these monkeys in the wild? What was their age in the captive sampling? When does the eruption of the third molar occur? What about, once again, the importance and pattern of allometric change? What about the possible differences in age expectancy between the wild and the captive specimens? This issue seems all the more important to be clarified than the pattern of shape change related to captivity (Figure 6) involves a massive change in the relative importance of the molar row compared to the rest of the mandible. Since molars do not remodel once erupted, a massive plastic change in the area of the molar row seems unlikely; the observed change is thus mostly an extension of the ramus, which develops with age. These anatomical aspects have to be discussed more thoroughly, together with the issue of the age distribution in the two groups, before jumping to an interesting but hypothetical interpretation.

In more details:

The introduction is well documented but it could contain more details about response to food consistency and remodeling along the animal's life, presumably important aspects of the variation in the wild, and potentially relevant to the difference between captive and wild animals. The approach to test the angles between vectors of shape change is smart; more references on similar approaches comparing vectors of shape changes may be welcome. However, testing the observed angle vs the null hypothesis of complete independence (90 degrees) may provide somewhat optimistic results. It would be more relevant to test whether the angles are actually correlated: a value of 67° seems quite a lot to state that the two directions are related. See for

instance the approach developed by D. Adams and implemented in the package geomorph. The number of young vs adult specimens has to be mentioned in the Table 1. Tables 3 and 4 have to be merged to improve readability. Please supply legends for the supplementary figures.

Decision letter (RSOS-181382.R0)

17-Dec-2018

Dear Dr Ito,

The editors assigned to your paper ("Phenotypic plasticity in the mandibular morphology of Japanese macaques: captive-wild comparison") have now received comments from reviewers. We would like you to revise your paper in accordance with the referee and Associate Editor suggestions which can be found below (not including confidential reports to the Editor). Please note this decision does not guarantee eventual acceptance.

Please submit a copy of your revised paper before 09-Jan-2019. Please note that the revision deadline will expire at 00.00am on this date. If we do not hear from you within this time then it will be assumed that the paper has been withdrawn. In exceptional circumstances, extensions may be possible if agreed with the Editorial Office in advance. We do not allow multiple rounds of revision so we urge you to make every effort to fully address all of the comments at this stage. If deemed necessary by the Editors, your manuscript will be sent back to one or more of the original reviewers for assessment. If the original reviewers are not available, we may invite new reviewers.

- Data accessibility

It is a condition of publication that all supporting data are made available either as supplementary information or preferably in a suitable permanent repository. The data

accessibility section should state where the article's supporting data can be accessed. This section should also include details, where possible of where to access other relevant research materials such as statistical tools, protocols, software etc can be accessed. If the data have been deposited in an external repository this section should list the database, accession number and link to the DOI for all data from the article that have been made publicly available. Data sets that have been deposited in an external repository and have a DOI should also be appropriately cited in the manuscript and included in the reference list.

If you wish to submit your supporting data or code to Dryad (<http://datadryad.org/>), or modify your current submission to dryad, please use the following link:
<http://datadryad.org/submit?journalID=RSOS&manu=RSOS-181382>

- **Competing interests**

- **Authors' contributions**

- **Acknowledgements**

- **Funding statement**

on behalf of Dr Ryan Earley (Associate Editor) and Kevin Padian (Subject Editor)
openscience@royalsociety.org

Comments to Author:

Reviewers' Comments to Author:

Reviewer: 1

Comments to the Author(s)

The paper is a sound, statistically good paper blessed with a good dataset. Although my general opinion is good, I'm worried by the excessive confidence of the authors in interpreting some results. I therefore suggest the authors to tone down some such insights, since they appear standing on thin ice at the moment.

More to the point:

1 You simply cannot control for genetic proximity in wild-caught individuals. How many of them a young males? do Japanese macaques distribute philopatrically? were sick or old individuals easier to collect? is there any pregnant female in the wild-caught sample? Please notice this matters to the evaluation of ecogeographical determinants of body size and body shape variation. Your result is perfectly reasonable, but all too many potential factors are simply ignored

2 at reading the intro I had the impression that sociality would be carefully scored and evaluated, but neither of the two. I wonder whether sociality fits anywhere in this manuscript, since it is either ignored (wild-caughts) or unnatural (captives)

3 the feeding status of captive individuals might be known, but did it vary seasonally? in wild individuals outside the tropics seasonal variation is the norm

4 being captive probably might elicit reaction norms of some sort, but most phenotypic variation is known to occur with domestic individuals, not wild-caught or F4 offspring of wild-caughts, so what is the basic premise of the ms? You have nice data there and a good treatment, and a nice paper. But I wonder if a better paper might be produced relaxing the basic expectations and premises (in the intro sections) and narrowing the scope (less sociality, less ecogeography, even less diet). I'd rather preserve these factors for the discussion, and produce a skinner, more frank presentation.

Thanks for the opportunity to go through this!

Pas

Reviewer: 2

Comments to the Author(s)

The manuscript deals with an interesting question, namely the impact of captive lifestyle on the facial morphology of wild animals, here Japanese macaques. The 3D morphometric approach is up-to-date and appropriate. Based on their analyses, the authors conclude that captive lifestyle had an important impact on facial shape, presumably because of hormonal changes in conditions where inter-individual relationships are departing from the wild. This may be an interesting result that would anyway deserve to be more thoroughly discussed in the background of what is known in captive representatives of other species (and that is clearly and with extensive citations presented in the introduction). Unfortunately, some drawbacks limit, in the present form, the support of the data to this interpretation.

First, a basic presentation of the results, with a PCA, a bgPCA or a CVA, and a plot of the size across the different populations, would help a lot the reader to get into the data. It would for instance show rather immediately the amount of variation related to the captive lifestyle compared to the variation in the wild. What is the amount of allometry in the shape variation? Second, the analyses with the Bayesian models seem quite sophisticated, but they may suffer of problems in their design. Mandible change with food consistency has been widely documented, and it seems rather surprising that the authors do not find any such trend across the wild

populations. What if they analyze wild populations alone? And if they analyze size-corrected data? Their sampling further includes small islets, where an insular syndrome could develop. They do not mention such potential sources of variations: as a basis for all other analyses, this has to be clarify.

Even more importantly, the authors only mention to have assessed the age issue by separating young (third molar not erupted) and adults (third molar erupted). They do not state when this event occurs in the animal's life. It seems anyway a quite basic way to deal with age, in a species which is presumably rather long lived. Bone remodeling in late life can be quite extensive, and occur following other directions than the changes occurring in early life (see Swiderski & Zelditch 2013). Therefore, more details and a better control for age is an absolute requirement for assessing the validity of all results and interpretations: what is the life expectancy of these monkeys in the wild? What was their age in the captive sampling? When does the eruption of the third molar occur? What about, once again, the importance and pattern of allometric change? What about the possible differences in age expectancy between the wild and the captive specimens? This issue seems all the more important to be clarified than the pattern of shape change related to captivity (Figure 6) involves a massive change in the relative importance of the molar row compared to the rest of the mandible. Since molars do not remodel once erupted, a massive plastic change in the area of the molar row seems unlikely; the observed change is thus mostly an extension of the ramus, which develops with age. These anatomical aspects have to be discussed more thoroughly, together with the issue of the age distribution in the two groups, before jumping to an interesting but hypothetical interpretation.

In more details:

The introduction is well documented but it could contain more details about response to food consistency and remodeling along the animal's life, presumably important aspects of the variation in the wild, and potentially relevant to the difference between captive and wild animals. The approach to test the angles between vectors of shape change is smart; more references on similar approaches comparing vectors of shape changes may be welcome. However, testing the observed angle vs the null hypothesis of complete independence (90 degrees) may provide somewhat optimistic results. It would be more relevant to test whether the angles are actually correlated: a value of 67° seems quite a lot to state that the two directions are related. See for instance the approach developed by D. Adams and implemented in the package geomorph. The number of young vs adult specimens has to be mentioned in the Table 1. Tables 3 and 4 have to be merged to improve readability.

Please supply legends for the supplementary figures.

Author's Response to Decision Letter for (RSOS-181382.R0)

See Appendix A.

RSOS-181382.R1 (Revision)

Review form: Reviewer 1 (Pasquale Raia)

Is the manuscript scientifically sound in its present form?

Yes

Are the interpretations and conclusions justified by the results?

Yes

Is the language acceptable?

Yes

Is it clear how to access all supporting data?

Yes

Do you have any ethical concerns with this paper?

No

Have you any concerns about statistical analyses in this paper?

No

Recommendation?

Accept with minor revision (please list in comments)

Comments to the Author(s)

The authors downplayed the excessive confidence in their data and frankly admit there are factors they were not in the position to control that could be important. Some few statements, especially in the closing lines, are still at odds with their data and results, and the language deserves further attention. I have nonetheless annotated the ms throughout to make this easy. I wish the author will appreciate this effort.

Regards

Pas

Decision letter (RSOS-181382.R1)

14-May-2019

Dear Dr Ito:

On behalf of the Editors, I am pleased to inform you that your Manuscript RSOS-181382.R1 entitled "Phenotypic plasticity in the mandibular morphology of Japanese macaques: captive-wild comparison" has been accepted for publication in Royal Society Open Science subject to minor revision in accordance with the referee suggestions. Please find the referees' comments at the end of this email.

The reviewers and Subject Editor have recommended publication, but also suggest some minor revisions to your manuscript. Therefore, I invite you to respond to the comments and revise your manuscript.

- Ethics statement

- Data accessibility

If you wish to submit your supporting data or code to Dryad (<http://datadryad.org/>), or modify your current submission to dryad, please use the following link:
<http://datadryad.org/submit?journalID=RSOS&manu=RSOS-181382.R1>

- Competing interests

- Authors' contributions

- Acknowledgements

- Funding statement

Because the schedule for publication is very tight, it is a condition of publication that you submit the revised version of your manuscript before 23-May-2019. Please note that the revision deadline will expire at 00.00am on this date. If you do not think you will be able to meet this date please let me know immediately.

on behalf of Dr Ryan Earley (Associate Editor) and Kevin Padian (Subject Editor)
openscience@royalsociety.org

Reviewer comments to Author:

Reviewer: 1

Comments to the Author(s)

The authors downplayed the excessive confidence in their data and frankly admit there are factors they were not in the position to control that could be important. Some few statements, especially in the closing lines, are still at odds with their data and results, and the language deserves further attention. I have nonetheless annotated the ms throughout to make this easy. I wish the author will appreciate this effort.

Regards

Pas

Author's Response to Decision Letter for (RSOS-181382.R0)

See Appendix B.

Decision letter (RSOS-181382.R2)

28-May-2019

Dear Dr Ito,

I am pleased to inform you that your manuscript entitled "Phenotypic plasticity in the mandibular morphology of Japanese macaques: captive-wild comparison" is now accepted for publication in Royal Society Open Science.

Kind regards,

on behalf of Dr Ryan Earley (Associate Editor) and Kevin Padian (Subject Editor)

Associate Editor Comments to Author (Dr Ryan Earley):

Associate Editor: 1

Comments to the Author:

(There are no comments.)

Reviewer comments to Author:

Appendix A

January 3rd, 2019

Dear editors and reviewers,

We are grateful that you have carefully read and provided favorable and constructive comments on the previous version of this manuscript. We carefully considered and addressed all the reviewers' comments. Mainly, both reviewers suggested that we should carefully interpret the results of wild–captive difference. In accordance with their suggestions, we mentioned the limitation of this study and discussed more about other sources of variations. We also performed additional analyses that consider the effects of age, island, and size. Overall, the comments helped us to expand cautious discussion, better highlight the validity of results, and to greatly improve the clarity of the text. The reply to individual comments begins on the next page.

We hope these revisions meet with your approval and look forward to receiving your response.

Yours Sincerely,

Tsuyoshi Ito,
on behalf of the co-authors

Reviewer: 1

Comments to the Author(s)

The paper is a sound, statistically good paper blessed with a good dataset. Although my general opinion is good, I'm worried by the excessive confidence of the authors in interpreting some results. I therefore suggest the authors to tone down some such insights, since they appear standing on thin ice at the moment.

REPLY: Thank you very much for your favorable and constructive comments to the manuscript.

More to the point:

1 You simply cannot control for genetic proximity in wild-caught individuals. How many of them are young males? do Japanese macaques distribute philopatrically? were sick or old individuals easier to collect? is there any pregnant female in the wild-caught sample? Please notice this matters to the evaluation of ecogeographical determinants of body size and body shape variation. Your result is perfectly reasonable, but all too many potential factors are simply ignored

REPLY: We agree that there are several unobserved factors that potentially impacts on the mandibular morphology. In answering your question, of 80 males, nine (six captive and three wild) are young individuals (we added the column showing this information in Table 1). In Japanese macaques, females usually stay the group of birth but males move to another group; therefore, adjacent populations are probably admixed due to male-induced gene flow. Unfortunately, there is no available data about the points you questioned. In accordance with your suggestions, we discussed such potentially-confounding factors and carefully interpreted the results (lines 348–368).

2 at reading the intro I had the impression that sociality would be carefully scored and evaluated, but neither of the two. I wonder whether sociality fits anywhere in this manuscript, since it is either ignored (wild-caughts) or unnatural (captives)

REPLY: As you questioned, we did not score or evaluate sociality but just discussed the potential influence of social hormone disturbance on mandibular morphology based on the captive–wild comparison. To avoid a misreading, we revised introduction to mention that the present study just compares wild and captive specimens (lines 108–118).

3 the feeding status of captive individuals might be known, but did it vary seasonally? in wild individuals outside the tropics seasonal variation is the norm

REPLY: The captive individuals in PRI had usually been fed monkey chow and sweet potato throughout the year. There is no systematic seasonal variation about diet.

4 being captive probably might elicit reaction norms of some sort, but most phenotypic variation is known to occur with domestic individuals, not wild-caught or F4 offspring of wild-caughts, so what is the basic premise of the ms? You have nice data there and a good treatment, and a nice paper. But I wonder if a better paper might be produced relaxing the basic expectations and premises (in the intro sections) and narrowing the scope (less sociality, less ecogeography, even less diet). I'd rather preserve these factors for the discussion, and produce a skinner, more frank presentation.

Thanks for the opportunity to go through this!

Pas

REPLY: In accordance with this suggestion, we refrained from mentioning about hypothesis in introduction but enhanced discussion by mentioning other potential sources of variations (lines 348–368). As for domestication, our basic premise is that, the morphological difference in captive individuals is mostly caused by environmental (plastic) rather than genetic factors, because our sample consists of at the maximum only four generations. In the revised manuscript, we also demonstrated that there was no shape change among generations except for founders (Table S1).

Reviewer: 2

Comments to the Author(s)

The manuscript deals with an interesting question, namely the impact of captive lifestyle on the facial morphology of wild animals, here Japanese macaques. The 3D morphometric approach is up-to-date and appropriate. Based on their analyses, the authors conclude that captive lifestyle had an important impact on facial shape, presumably because of hormonal changes in conditions where inter-individual relationships are departing from the wild. This may be an interesting result that would anyway deserve to be more thoroughly discussed in the background of what is known in captive representatives of other species (and that is clearly and with extensive citations presented in the introduction). Unfortunately, some drawbacks limit, in the present form, the support of the data to this interpretation.

REPLY: Thank you very much for your favorable and constructive comments to the manuscript.

First, a basic presentation of the results, with a PCA, a bgPCA or a CVA, and a plot of the size across the different populations, would help a lot the reader to get into the data. It would for instance show rather immediately the amount of variation related to the captive lifestyle compared to the variation in the wild. What is the amount of allometry in the shape variation?

REPLY: In accordance with this suggestion, we showed the plot of PCA and size across populations, which is color-coded by environment (captive vs wild) (Fig. S3). As for allometry, shape (Procrustes coordinates) was slightly but significantly explained by size ($R^2 = 0.023$, $P = 0.048$) (lines 212–215).

Second, the analyses with the Bayesian models seem quite sophisticated, but they may suffer of problems in their design. Mandible change with food consistency has been widely documented, and it seems rather surprising that the authors do not find any such

trend across the wild populations. What if they analyze wild populations alone? And if they analyze size-corrected data? Their sampling further includes small islets, where an insular syndrome could develop. They do not mention such potential sources of variations: as a basis for all other analyses, this has to be clarify.

REPLY: We agree that size (allometry) and island effect are potential sources of variations. We had evaluated the effect of allometry by incorporating size as explanatory factor. We also had performed analyses with wild populations alone and had found that there were no clear effects of ecogeographic factors (Fig. 8). In the revised manuscript, to make doubly sure about the control of allometric effect, we reproduced the analyses with size-adjusted shape data (residuals from the multiple regression of Procrustes coordinates on size); even in this case, no clear effect of ecogeographical factors was detected (Fig. S8; lines 271–276). We also reproduced the analyses with island as explanatory factor and found that temperature may possibly have an effect on mandibular shape (Fig. S9; lines 276–278). In accordance with the results of these additional analyses, we discussed the possibility of the influence of temperature on mandibular variation (lines 313–328).

Even more importantly, the authors only mention to have assessed the age issue by separating young (third molar not erupted) and adults (third molar erupted). They do not state when this event occurs in the animal's life. It seems anyway a quite basic way to deal with age, in a species which is presumably rather long lived. Bone remodeling in late life can be quite extensive, and occur following other directions than the changes occurring in early life (see Swiderski & Zelditch 2013). Therefore, more details and a better control for age is an absolute requirement for assessing the validity of all results and interpretations:

REPLY: This issue interests us, because the morphological change after maturation (third molar eruption) has often been overlooked in the studies of non-human primate skulls. We addressed this issue as follows.

what is the life expectancy of these monkeys in the wild?

REPLY: Little is known about the life expectancy in the wild. A handful of studies reported that the life span varies across populations; that of females (mean \pm SD) is 6.3 ± 5.2 (Yakushima, nonprovisioned natural population), 8.4 (Takasakiyama, provisioned natural population), and 13.6 ± 0.4 (Arashiyama, provisioned natural/semi-natural population) (Fooden and Aimi, 2005). The greatest accurately known life span in Japanese macaques are 33 years in a female in a provisioned natural group and 28 years in a male another provisioned natural group (Fooden and Aimi, 2005).

What was their age in the captive sampling?

REPLY: In the captive specimens of known ages, age ranges from 6.4 to 29.5 years (mean \pm SD: 14.0 ± 5.7) (lines 137–138).

When does the eruption of the third molar occur?

REPLY: In the captive specimens of known ages, the individuals with fully erupted third molars (adults) were observed in 8.1 years of age or more (lines 139–140). To rule out the variation between young adult and adult specimens, we reproduced the analyses with restricting the sample to the adults with fully-erupted third molars (lines 221–222). Even in this case, significant captive–wild difference was detected (Fig. S7; lines 258–259).

What about, once again, the importance and pattern of allometric change?

REPLY: To make doubly sure about the control of allometric effect, we reproduced the analyses with replacing the size-adjusted shape data with response variables (lines 211–215). Even in this case, significant captive–wild difference was detected (Fig. S4; lines 255–256).

What about the possible differences in age expectancy between the wild and the captive specimens?

REPLY: Common sense would say that captive individuals can possibly live longer on average than wild individuals (although it is not necessarily the case because the captive individuals of PRI are sometimes used for experimental research). Considering the reported life span in natural populations (see above comments), individuals older than 20 years of age are considered to be very rare in wild samples. To address this issue, we reproduced the analyses with removing the old-aged captive individuals (20 years of age or more) (lines 217–220). Even in this case, significant captive–wild difference was detected (Fig. S6; lines 257–258).

This issue seems all the more important to be clarified than the pattern of shape change related to captivity (Figure 6) involves a massive change in the relative importance of the molar row compared to the rest of the mandible. Since molars do not remodel once erupted, a massive plastic change in the area of the molar row seems unlikely; the observed change is thus mostly an extension of the ramus, which develops with age. These anatomical aspects have to be discussed more thoroughly, together with the issue of the age distribution in the two groups, before jumping to an interesting but hypothetical interpretation.

REPLY: We assume that the captive–wild difference in mandibular morphology was formed not after maturation (the eruption of third molars) but during development (particularly in adolescent). In fact, we showed that there is no significant shape change (in PC4 and shape score) after maturation (lines 259–261). In adolescence (three to five years of age), canines, premolars, second and third molars are not erupted or under eruption. As the reviewer #2 pointed out, it is unlikely that the size of each tooth changes. Therefore, we consider that the observed changes possibly reflect the remodeling of both mandibular body and ramus. In accordance with the reviewer#2's suggestion, we discussed more about the anatomical aspect of this change (lines 331–333) and about the issue of the age distribution (lines 349–361).

In more details:

The introduction is well documented but it could contain more details about response to food consistency and remodeling along the animal's life, presumably important aspects of the variation in the wild, and potentially relevant to the difference between captive and wild animals.

REPLY: In accordance with this comment, we added the description about response to food consistency (lines 63–64) and remodeling along animal's life (lines 356–360).

The approach to test the angles between vectors of shape change is smart; more references on similar approaches comparing vectors of shape changes may be welcome. However, testing the observed angle vs the null hypothesis of complete independence (90 degrees) may provide somewhat optimistic results. It would be more relevant to test whether the angles are actually correlated: a value of 67° seems quite a lot to state that the two directions are related. See for instance the approach developed by D. Adams and implemented in the package geomorph.

REPLY: In accordance with this comment, we cited reference that used similar approach comparing vectors (Pitchers et al. 2013). We also added the plot of correlations between vectors (Fig. 7b).

The number of young vs adult specimens has to be mentioned in the Table 1.

REPLY: In accordance with this comment, we mentioned the number of young adult and adult specimens in Table 1.

Tables 3 and 4 have to be merged to improve readability.

REPLY: We merged Tables 3 and 4 accordingly.

Please supply legends for the supplementary figures.

REPLY: We supply legends for the supplementary figures accordingly.

The comments annotated on the attached PDF:

is this relevant? just delete (on about the following sentence: “Because of this, Japanese macaques are also beginning to attract the attention of researchers as an analogy for cold 90 adaptation in humans [40].”)

REPLY: We deleted this sentence accordingly.

isn't ontogeny a big neglected factor here? (on about the following sentence: “The individuals with smaller PC scores (i.e. smaller individuals or males rather than females) tended to be squared-jawed (laterally positioned gonions) with a more posteriorly inclined ramus [Fig. 4(c)]. PC5–10 were not significantly explained by any explanatory factors.”)

REPLY: PC3 is not explained by age class, and therefore it is unlikely that this is ontogenetic variation.

how can you tell? (on about the following sentence: “In Japanese macaques, the nutritional conditions of captivity were unlikely to have been so markedly different from natural conditions as to have influenced the growth of the body or mandible.”)

REPLY: We recognized that this sentence was overgeneralized. In the revised manuscript, we mentioned only about size here (lines 287–290). We also corrected all the editorial issues annotated on the PDF.

Appendix B

May 20th, 2019

Dear editors and reviewers,

I am grateful that you have carefully read and provided favorable and constructive comments on the previous version of this manuscript. In accordance with the reviewer's comments, I revised the manuscript as follows. The reply to the comments begins on the next page.

I hope these revisions meet with your approval and look forward to receiving your response.

Yours Sincerely,

Tsuyoshi Ito,
on behalf of the co-authors

Reviewer: 1

Comments to the Author(s)

The authors downplayed the excessive confidence in their data and frankly admit there are factors they were not in the position to control that could be important. Some few statements, especially in the closing lines, are still at odds with their data and results, and the language deserves further attention. I have nonetheless annotated the ms throughout to make this easy. I wish the author will appreciate this effort.

Author's response: Thank you very much for kindly annotating our manuscript throughout. This was very helpful to make the text clear. I revised the manuscript in accordance with all the edits. The reviewer also made three questions. The answers to these are as follows.

The reviewer's question: do you mean from ten different populations in the wild?

Author's response: The individuals came from ten different natural provisioned or non-provisioned populations. In the revised manuscript, I added the text explaining this (line 123).

The reviewer's question: it 8.1 is the age of the youngest with M3, how comes your sample of captives starts at 6.4?

Author's response: Adults with fully erupted molars are 8.1 years old or more. The samples of young adults with almost erupted molars include the individuals with less than 8.1 years old (e.g. 6.4 years old individual).

The reviewer's question: please check, it looks like females are the larger sex...

Author's response: As the reviewer indicated, this is confusing result. I consider that this may be the consequence of multicollinearity between size and sex, because the two variables are highly correlated to each other ($r = 0.75$) and because, when removing one of the two variables, the effect of the other became insignificant (Figures S4 and S5). I revised the text to make this clear (lines 231–234, 305–306). In related to this, I noticed that the shape visualizations along principal components in Figure 4 were in opposite direction. I fixed Figure 4 and accordingly revised the relevant parts of the text (lines 225–226). This does not affect our conclusion.